# Sex-specific association between gut microbiome and fat distribution

Yan Min [1,2,6], Xiaoguang Ma[3,4,6], Kris Sankaran[5], Yuan Ru[3,4], Lijin Chen[3,4], Mike Baiocchi[1,2,5] & Shankuan Zhu[3,4]

The gut microbiome has been linked to host obesity; however, sex-specific associations between microbiome and fat distribution are not well understood. Here we show sex-specific microbiome signatures contributing to obesity despite both sexes having similar gut micro-biome characteristics, including overall abundance and diversity. Our comparisons of the taxa associated with the android fat ratio in men and women found that there is no widespread species-level overlap. We did observe overlap between the sexes at the genus and family levels in the gut microbiome, such as *Holdemanella* and *Gemmiger*; however, they had opposite correlations with fat distribution in men and women. Our findings support a role for fat distribution in sex-specific relationships with the composition of the microbiome. Our results suggest that studies of the gut microbiome and abdominal obesity-related disease outcomes should account for sex-specific differences.

[1] Stanford Prevention Research Center, Stanford School of Medicine, Stanford CA 94305, USA. [2] Department of Health Research and Policy (Epidemiology), Stanford School of Medicine, Stanford CA 94305, USA. [3] Department of Nutrition and Food Hygiene, School of Public Health, Zhejiang University, Hangzhou 310058, China. [4] Chronic Disease Research Institute, School of Public Health, and Women's Hospital, Zhejiang University School of Medicine, 310058 Hangzhou, Zhejiang, China. [5] Department of Statistics, Stanford University, Stanford CA 94305, USA. [6]These authors contributed equally: Yan Min, Xiaoguang Ma. Correspondence and requests for materials should be addressed to M.B. (email: baiocchi@stanford.edu) or to S.Z. (email: zsk@zju.edu.cn)

The human gut microbiome is a complicated, dynamic community consisting of 10–100 trillion microbes, which carries ~100 times more genes than the human genome[1,2]. Studies have associated the gut microbiome with the host obesity, however, little is known about how does sex modulate the association between gut microbiome and fat distribution. As men on average have lower percentages of total body fat but are more susceptible to abdominal adiposity than women[3–5], we consider sex as a pivotal factor to further the understanding of the relationship between the gut microbiome and fat distribution. This may link to men's elevated risk of having abdominal obesity-induced adverse health outcomes[4,6,7].

Studies revealed that the microbial transplantation from obese to lean mice resulted in significant weight gain[8,9]. The findings suggested certain microbes were directly responsible for the host obesity and enabled the weight gain in lean mice. Studies involving human subjects also indicated that the obese and the lean adults had different microbiota properties: their microbial communities had different dominant phyla; their microbial signature could be composed of different species; and the microbiome diversity appeared to be lower among the obese subjects compared to the lean individuals[10–12]. Further, studies also hypothesized that the gut microbiota can potentially differ in men and women, due to the influence of the overall obesity[13]. These observations in both animal and human studies confirmed the association between the gut microbiome and obesity. Moreover, fat distribution, independent from total adiposity, has been well recognized as a major predictor of cardiovascular and metabolic outcomes[14,15]. Our study concerns the sex-specific associations between the gut microbiome and abdominal obesity among the natural population using high-precision fat distribution measurements.

In this study, we hypothesize that the gut microbiome is not only associated with the overall obesity, but also the fat distribution. Since men and women differ in both total body fat proportion and distribution, we further hypothesize that there are sex-specific microbiome signatures associated with the fat distribution in men and women. Such microbial signatures have not been investigated to date. To test these hypotheses, we adopted body composition measurements from dual x-ray absorptiometry (DXA), and the gut microbiome information from 16s rRNA sequencing. we first visualized the unadjusted association between the android fat ratio and microbiome characteristics, including its abundance and diversity, the gynoid fat ratio and the same microbiome characteristics. We then conducted microbiome-wide taxa level association testing to identify the taxa that are associated with men and women. The results show similar microbiome characteristics, including overall abundance and diversity, in relation to fat distribution in both sexes; however, there is no wide-spread species-level overlap among those microbial taxa associated with fat distribution in men and women. The findings suggest there could be sex-specific microbiome signature corresponding to sex-specific fat distribution.

## Results

**Demographic characteristics**. The participants of this microbiome study came from the same sub-district, different households of the WELL-China study. Ten individuals did not have DXA assessment, did not provide stool samples, or had missing values on covariates were excluded from the study, leaving 212 for the final analysis. Demographic characteristics of these 212 subjects were summarized in Table 1. Of these 212 subjects, 45% were men and 55% were women. The overall mean age is 51 years for both sexes. About 61% of men versus almost 0% of women

were current smokers; 70% of men versus 30% women are current alcohol drinkers. The prevalence for metabolic syndrome was 17% in men and 16% in women; the prevalence for type 2 diabetes was 20% in men and 11% in women. Comparing the male and the female participants using $t$-tests for continuous variables and chi-square tests for categorical variables, the following characteristics are statistically significant at the $p = 0.005$ level: education, waist-to-hip ratio, android fat ratio, gynoid fat ratio, daily carbohydrate intake, current smoker, current alcohol drinker, and high-density lipoprotein.

**Global association by sex**. For each sex, samples were divided into four quartiles according to the measured android and gynoid fat ratio. The associated intervals of female android and gynoid fat ratio were (6.6%, 9.0%), (9.0%, 10.0%), (10.0%, 10.8%), and (10.8%, 13.8%), and (12.4%, 15.6%), (15.6%, 17.0%), (17.0%, 19.6%), and (19.6%, 26.6%), respectively. The intervals of male android and gynoid fat ratio were (9.1%, 11.6%), (11.6%, 12.6%), (12.6%, 13.35), and (13.3%, 15.4%), and (11.3%, 13.8%), (13.8%, 15.5%), (15.5%, 17.1%), and (17.1%, 25.0%), respectively. The microbiome tertiles were created using the sum of the taxon abundance of each subject. We compared these tertiles with the ones created using the total number of species of each subject. The two tertiles are very consistent. Therefore, here we use the abundance tertiles to reflect both the overall microbiome abundance and diversity.

Stratified by sex, four heat maps were created to summarize the unadjusted association between microbiome tertiles and android or gynoid fat ratio. The top bar color-coded the fat ratio quartiles in male and female participants. From light to dark blue, it represented the lowest to the highest quartiles in each sex sequentially. The second bar is the microbiome tertiles, with the first tertile as the least diverse and abundant group, the third as the highest. Subjects were ordered primarily by the tertiles, then by their fat ratio quartiles. Thus, the two bars together showed how did the four fat ratio quartiles form each microbiome tertile. The rows of the heat maps were the top 50 abundant species ranked high-to-low from bottom up, and the color gradient from white to red indicated the transformed abundance of every species for each subject.

Figure 1a shows the association of microbiome abundance and the android fat ratio in female subjects. 36% of the subjects in the lowest abundance tertile are from the highest android fat ratio quartile, whereas the second and third tertiles only contained 24% and 15%, respectively. Although results of Fisher's exact test ($p = 0.19$) and the univariate linear regression ($p = 0.06$) were not significant, Fig. 1a did imply that as the microbiome abundance and diversity increased, the ratio of high android fat individuals decreased, and the ratio of low android fat individuals increased. As shown in Fig. 1b the second microbiome tertile had the most subjects from the highest android fat ratio quartile. The result from the Fisher's exact test ($p = 0.81$) was not significant. In the linear regression, the main effect of android fat ($p = 0.41$), the quadratic effect ($p = 0.28$) were not significant.

Figure 1c, d shifted the attention from android fat ratio to gynoid fat ratio in the same setup. In the female subjects, the second microbiome tertile has the least subjects from the highest android fat ratio group. The results from both Fisher's Exact test ($p = 0.19$) and the linear regression ($p = 0.19$ for android fat ratio, $p = 0.29$ for the quadratic term of android fat ratio) were not significant. However, in male subjects, there was a clear increasing trend in gynoid fat ratio as the microbiome abundance and diversity increased. 25% of the subjects in the lowest microbiome abundance tertile were from the highest gynoid fat ratio quartile; 42% in the third tertile. Although the results of

**Table 1 Demographic characteristics**

|  | Male, *N* (%) | Female, *N* (%) | *p*-Values[a] |
|---|---|---|---|
| Total (*N* = 212) | 96 (45%) | 116 (55%) |  |
| Age (years), mean (SD) | 50.7 (14.5) | 50.7 (14.1) |  |
| *Marital status* |  |  | 0.502 |
|   Single | 5 (5.2) | 5 (4.3) |  |
|   Married | 89 (91.7) | 102 (93.1) |  |
|   Widowed | 1 (1.0) | 3 (2.6) |  |
|   Divorced | 2 (2.1) | 0 (0) |  |
| *Education* |  |  | <0.005 |
|   Illiterate | 15 (15.5) | 40 (34.5) |  |
|   Elementary school | 19 (19.6) | 28 (24.3) |  |
|   Middle school | 30 (30.9) | 14 (12.1) |  |
|   High school | 19 (19.6) | 17 (14.6) |  |
|   Junior college, college, & above | 14 (14.4) | 17 (14.5) |  |
| *Anthropometrics* |  |  |  |
| Body mass index, mean (SD) | 23.6 (3.0) | 23 (3.0) | 0.185 |
| Waist-to-hip ratio, mean (SD) | 0.94 (0.07) | 0.89 (0.08) | <0.005 |
| Android fat ratio, mean (SD) | 12.5 (1.2) | 9.9 (1.4) | <0.005 |
| Gynoid fat ratio, mean (SD) | 15.9 (3.0) | 17.7 (3.0) | <0.005 |
| *Life style behaviors* |  |  |  |
| Daily carbohydrate intake (g), median (IQR) | 223.2 (114.6) | 182.0 (74.6) | <0.005 |
| Daily fat intake (g), median (IQR) | 29.6 (29.4) | 30.7 (29.3) | 0.999 |
| Current smoker | 71 (60.7) | 0 (0) | <0.005 |
| Current alcohol drinker | 81 (69.8) | 38(29.9) | <0.005 |
| Antibiotics use (past 3 months) | 15 (15.6) | 15 (12.9) | 0.717 |
| *Disease history* |  |  |  |
| Metabolic syndrome[b] | 16 (16.7) | 19 (16.4) | 0.999 |
| Type 2 diabetes[c] | 13 (20.0) | 9 (10.6) | 0.167 |
| Hypertension[d] | 19 (19.8) | 15 (13.0) | 0.243 |
| *Biomarkers* |  |  |  |
| Fasting blood glucose (mmol/L), mean (SD) | 5.4 (1.6) | 5.2 (0.9) | 0.190 |
| Insulin (pmol/L), mean (SD) | 40.6 (22.0) | 47.9 (20.5) | 0.020 |
| Total cholesterol (mmol/L), mean (SD) | 5.0 (0.9) | 5.0 (0.8) | 0.992 |
| Triglycerides (mmol/L), mean (SD) | 1.7 (1.2) | 1.4 (1.5) | 0.146 |
| Low-density lipoprotein (mmol/L), mean (SD) | 2.8 (0.7) | 2.7 (0.7) | 0.390 |
| Hight-density lipoprotein (mmol/L), mean (SD) | 1.5 (0.4) | 1.7 (0.4) | <0.005 |

[a]*p*-values are from chi-square tests for categorical variables and *t* tests for continuous variables by comparing the characteristics between men and women
[b]Metabolic syndrome was defined using International Diabetes Federation criteria for the Chinese population, all the measurements were taken at the baseline at the same time the stool samples were collected
[c]Type 2 diabetes was defined as HbA1c ≥ 6.4%
[d]Hypertension was defined as systolic blood pressure ≥ 140 mmHg or diastolic blood pressure ≥ 90 mmHg

Fisher's exact test ($p = 0.29$) and the univariate linear regression ($p = 0.09$) were not significant, it was still worth to observe the differences across the microbiome abundance tertiles in Fig. 1d.

**Sex-specific microbiome and android fat ratio.** Wald tests were performed on 336 taxa in the female samples and 324 in the male samples. These two models tested android fat ratio as the major exposure at two-sided significance value of 0.01 in male and female subjects. Only those significant results with Log2 fold change ≥ 1 or ≤−1 were reported as final results in Table 2. Log2 fold change ≥ 1 indicates that taxa abundance was at least doubled with 1 standard deviation unit change in the android fat ratio; Log2 fold change ≤ −1 indicated the taxa abundance was at least reduced in half with 1 standard deviation unit change in the android fat ratio. The standard deviations of android fat ratio were 1.2%, 1.4% for male and female, respectively.

Table 2 summarized the results by the direction of the association stratified by sex. The first column was the taxa ID, which was a unique ID generated to differentiate different species within the same genus. The second column provided the family level information, the third column indicated the genus. The dictionary mapping taxa ID back to the original sequences was provided in the Supplementary Source Data. The rows banded in the same color across Tables 2 and 3 indicated they were exactly the same sequence of one taxon.

In the female sample, the model did not provide enough evidence of any taxon that was positively associated with android fat ratio. Four taxa from three families in male subjects resulted having positive associations with android fat ratio. *Bacteroides* from *Bacteroidaceae* family and *Holdemanella* from *Erysipelotrichaceae* family had effect sizes larger than 9. This indicated with one-standard deviation increase of android fat ratio, these two taxa abundance increased at least $2^9$ times. Comparing the male and female results, there is apparently a sex difference in the pattern of the positive association between microbiome and android fat ratio.

One taxon from the *Erysipelotrichaceae*, *Holdemanella* genus showed significant negative association with android fat ratio in the female sample. Notably, *Holdemanella* genus was both positively associated with android fat ratio in male and negatively associated in female, however, two different species from this genus were responsible for the observed effects. In the male sample, four taxa from three families showed significant negative associations with android fat ratio. Interestingly, a taxon also from *Bacteroidaceae* family, the *Bacteroides* genus had the largest effect size of −7.5. Among the male and female taxa that were negatively associated with android fat ratio, no overlap at either genus or family level was observed.

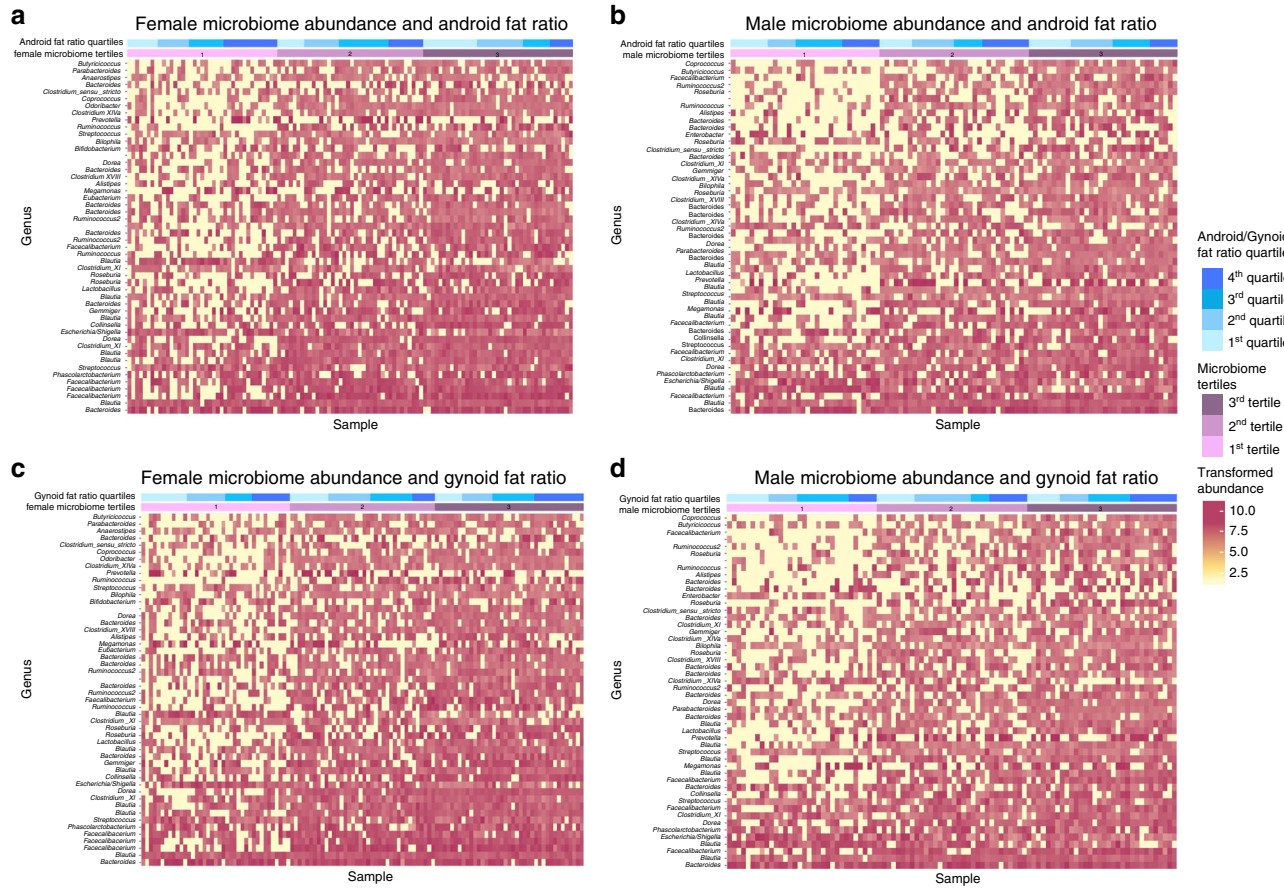

**Fig. 1** The association of microbiome abundance and fat ratio. This figure shows the global relationship between microbiome abundance and fat distribution in men and women. The top bar indicates the distribution of android and gynoid fat ratios within each microbiome abundance tertile. The blue gradient from light to dark encode the quartiles of the fat ratio from the lowest to the highest. The second bar represents the microbiome abundance tertiles. From light to dark purple, it indicates the lowest to the highest microbiome abundance tertiles. The heatmap contains the top 50 most abundant microbial taxa in men and women ranked from the most abundant to the least bottom up. They compose the rows of the heatmap. Each column represents one sample. The color gradient indicates the abundance of a microbial taxa in a particular sample. Panels **a**, **b** show the relationships between microbiome abundance and android fat ratio in women and man, respectively. Similarly, panels **c**, **d** show the associations between microbiome abundance with gynoid fat ratio in women and men

**Table 2 Taxa associated with android fat ratio (full model[a])**

|          |      | Taxa ID | Family             | Genus           | Log2 fold change | P-adj[b] |
|----------|------|---------|--------------------|-----------------|------------------|----------|
| Positive | Male | ID. 114 | Bacteroidaceae     | Bacteroides     | 9.7              | 1.7E−05  |
|          |      | ID. 327 | Erysipelotrichaceae| Absiella        | 7.7              | 4.7E−05  |
|          |      | ID. 108 | Erysipelotrichaceae| Holdemanella    | 10.0             | 1.6E−05  |
|          |      | ID. 113 | Ruminococcaceae    | Gemmiger        | 7.7              | 5.1E−05  |
| Negative | F[c] | ID. 193 | Erysipelotrichaceae| Holdemanella    | −11.1            | 5.1E−06  |
|          | Male | ID. 225 | Bacteroidaceae     | Bacteroides     | −7.5             | 4.4E−03  |
|          |      | ID. 215 | Prevotellaceae     | Paraprevotella  | −7.1             | 8.4E−03  |
|          |      | ID. 75  | Ruminococcaceae    | Clostridium_IV  | −4.8             | 4.9E−03  |
|          |      | ID. 150 | Ruminococcaceae    | Gemmiger        | −7.2             | 1.3E−04  |

[a]Adjusted for age, BMI, smoking, alcohol use, dietary fat intake, dietary carbohydrate intake, total energy intake, antibiotic use, and sequencing batch
[b]p-Values from the Wald tests are adjusted by Benjamini–Hochberg method
[c]F here stands for female

**Sex-specific microbiome and gynoid fat ratio.** Wald tests were also performed to test gynoid fat ratio and its associated taxa stratified by sex. The same criteria were applied in reporting the results. The standard deviations of gynoid fat ratio were both 3% in the male and female.

In the female sample, three taxa from two families were discovered having significant positive association with gynoid fat

ratio. The taxon from *Prevotellaceae* family, *Prevotella* genus showed the highest effect size of 9.6. A taxon from *Lachnospiraceae* family, *Clostridium_XlVa* genus showed a positive association with effect size of 10.2 in the male sample.

Three taxa from three families in the female sample were negatively associated with gynoid fat ratio, with the largest effect size of −10.9 coming from a taxon of the *Rikenellaceae* family,

*Alistipes* genus. In the male sample, there were four taxa from three families having significant negative associations with gynoid fat ratio. The taxon from the *Bacteroidaceae* family, *Bacteroides* genus had the largest effect size of −24.2. Comparing male and female results associated with gynoid fat ratio, there was no genus level overlap with respect to the same association. However, *Gemmiger* genus from *Ruminococcaceae* family had a positive association with gynoid fat ratio in the female sample, a negative association in the male sample. Since the two taxa had the same taxa ID, the two were exactly the same species but had completely opposite effects in women and men.

## Discussion

Results from the global association indicated that men and women shared similar association of android fat ratio and microbiome abundance. Although Fisher's exact test and the univariate linear regression produced insignificant results, this could be due to the small sample size. The heat maps (Fig. 1a, b) suggested that as the microbiome abundance and diversity increased, the android fat ratio decreased in both sexes. Moreover, the heat maps (Fig. 1c, d) suggested a potential negative association between gynoid fat ratio and microbiome abundance in both sexes. Even though, at this aggregated microbiome abundance level, the relationship between fat distribution and microbiome abundance in two sex were similar, when zoomed in to the taxa level, we discovered two different sets of microbiomes that were responsible for this relationship.

Figure 2 summarizes all the taxa that were significantly associated with either android or gynoid fat ratio in by sex. The color coding indicated the type (android or gynoid) and the direction (positive or negative) of the association. The height of the bar indicated the absolute value of the effect size. Each segment of the inner circle represented one bacteria family.

Figure 2a shows a total of seven taxa that showed significant associations with either android or gynoid fat ratio in the

### Table 3 Taxa associated with gynoid fat ratio (full model[a])

|  |  | Taxa ID | Family | Genus | Log2 fold change | *P*-adj[b] |
|---|---|---|---|---|---|---|
| Positive | Female | ID. 180 | Prevotellaceae | Paraprevotella | 7.9 | 2.3E−03 |
|  |  | ID. 59 | Prevotellaceae | Prevotella | 9.6 | 9.9E−03 |
|  |  | ID. 113 | Ruminococcaceae | Gemmiger | 8.2 | 6.5E−03 |
|  | M | ID. 524 | Lachnospiraceae | Clostridium_XIVa | 10.2 | 4.7E−03 |
| Negative association | Female | ID. 271 | Lactobacillaceae | Lactobacillus | −6.6 | 6.5E−03 |
|  |  | ID. 294 | Rikenellaceae | Alistipes | −10.9 | 6.5E−03 |
|  |  | ID. 187 | Ruminococcaceae | Ruminococcus | −9.1 | 1.9E−03 |
|  | Male | ID. 114 | Bacteroidaceae | Bacteroides | −24.2 | 1.7E−21 |
|  |  | ID. 214 | Lachnospiraceae | Clostridium_XIVa | −5.7 | 4.5E−03 |
|  |  | ID. 151 | Lachnospiraceae | Coprococcus | −10.0 | 5.5E−03 |
|  |  | ID. 113 | Ruminococcaceae | Gemmiger | −10.9 | 2.8E−03 |

[a]Adjusted for age, BMI, smoking, alcohol use, dietary fat intake, dietary carbohydrate intake, total energy intake, antibiotic use, and sequencing batch
[b]*p*-values from the Wald tests are adjusted by Benjamini–Hochberg method

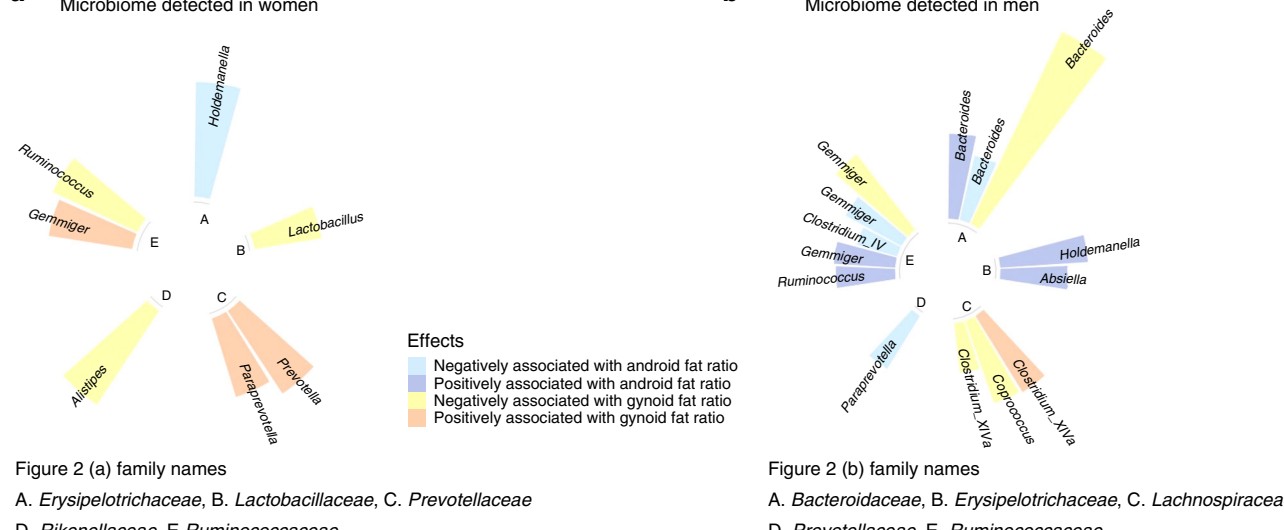

Figure 2 (a) family names
A. *Erysipelotrichaceae*, B. *Lactobacillaceae*, C. *Prevotellaceae*
D. *Rikenellaceae*, E. *Ruminococcaceae*

Figure 2 (b) family names
A. *Bacteroidaceae*, B. *Erysipelotrichaceae*, C. *Lachnospiraceae*
D. *Prevotellaceae*, E. *Ruminococcaceae*

**Fig. 2** Taxa associated with fat distribution in men and women. The figure summarizes four associations between microbiome abundance and fat distribution. The colors, from blue to red, encodes the following four effects, respectively: negative and positive associations with android fat ratio, negative and positive associations with gynoid fat ratio. The height of each bar indicates microbiome abundance on a log2 fold change scale. The inner circular segments separate microbial taxa by their family names. Panel **a** shows the four effects among female samples and panel **b** shows the effects in male samples

female sample. Although the android and gynoid fat were negatively associated with a Pearson correlation coefficient of −0.82 in women. The two taxa from *Ruminococcaceae* family: *Ruminococcus* and *Gemmiger* showed disagreed effects, as *Gemmiger* was positively associated with gynoid fat ratio, whereas *Ruminococcus* had a negative association. *Prevotellaceae* family had two taxa from the *Prevotella* and *Paraprevotella* genera that had the same effect—positively associated with gynoid fat ratio. The rest three families: *Erysipelotrichaceae*, *Lactobacillaceae*, and *Rikenellaceae* only had one taxon made to the final results. Similarly, Fig. 2b summarizes all the 14 taxa having significant associations with android or gynoid fat ratio in the male sample. The Pearson correlation coefficient of android fat ratio and gynoid fat ratio was −0.5 in men. Three taxa from *Bacteroidaceae* and five taxa *Ruminococcaceae* families covered three effects: both directions of the association with android fat ratio, and negative association with gynoid fat ratio. Three taxa from *Lachnospiraceae* family covered both effect directions with gynoid fat ratio. Two taxa from *Erysipelotrichaceae* family had the agreed effect. *Prevotellaceae* family's *Paraprevotella* genus only had one taxon made into the final result.

Comparing the results of the two sexes, there were three family-level overlaps: *Erysipelotrichaceae*, *Prevotellaceae*, and *Ruminococcaceae*. However, genus-level associations within these overlapped families did not always agree between men and women. Different taxa from *Holdemanella* genus, *Erysipelotrichaceae* family showed a negative association with android fat ratio in women, but a positive association in men. Similarly, taxa from *Gemmiger* genus from *Ruminococcaceae* family was positively associated with gynoid fat ratio in women, but both positively and negatively associated with android fat ratio in men. These results indicated that on one hand, men and women had different microbiome species associated with fat distribution; on the other hand, the same family and genus of microbiome could have different associations with fat distribution in the two sexes.

Comparing to other studies, Haro et al. reported *Bacteroides* genus in men was negatively associated with BMI, however no association was observed in women[13]. In our study, we did not observe any *Bacteroides* effect in women; however, in men, different taxa from the *Bacteroides* had both positive and negative effects on android fat ratio. Most of the studies did not stratify their findings by sex. Kasai et al. reported species from the *Ruminococcus* genus were significantly more prevalent in obese individuals versus the non-obese[11]; Million et al. pointed out that although previous studies claimed that *Lactobacillus* genus had a critical role in anti-obesity, difference species from the same genus could actually be positively associated with obesity[12]. Our results showed negative association between taxa from *Lactobacillus* and *Ruminococcus* genera and gynoid fat ratio in women, not in men. However, we have observed the same phenomenon such as taxa from the same genus presenting opposite effect in our studied population.

Previous studies have shown a positive association between testosterone and android adiposity, and a positive association between estrogen and gynoid fat deposition[4,16], which are likely to partially explain the difference in fat distribution between men and women. Besides the association between fat distribution and hormone, gut microbiota can also be affected by the systemic sex hormone level. Yurkovetskiy et al. compared the male and female microbiota before and after puberty in mouse models, concluded that puberty-related hormonal changes led to the separation in the male and female microbiota, as prepubescent differences were not observed. They also observed the microbiota of castrated postpubescent male mice getting similar to their female counterparts. Inversely, the study

also showed that microbiota elevated androgens in the mouse model[17]. It has also been reported that changes in sex hormone levels were associated with microbiome compositions in human[18,19]. Although the mechanism is still unclear, there seem to be a feedback loop between sex hormones and the gut microbiota. Further, gut microbiota also modulates fat deposition through producing short-chain fatty acid (SCFA) as a bacterial fermentation product, increasing enzyme lipoprotein lipase (LPL), which catalyzes the release of fatty acid, and stimulating local angiogenesis and vascular remodeling which induces local fat accumulation[20–23]. Evidence have shown obesity alters microbial colony in both mouse models and human studies, but no studies have addressed how does regional adiposity, in turn, affects local microbiota. Potentially, like the interaction between sex hormone and microbiome, there is also a feedback loop between regional adiposity and microbiome. According to our study, the taxa associated with android and gynoid fat ratio in men and women, although sharing moderate overlap at family and genus level, do not have any species-level overlap responsible for the same association. This indicates sex could potentially determine a subgroup of gut microbiota that are sex-hormone sensitive and responsible for regional adiposity. Due to the observed sexual dimorphism of the detected microbial taxa in men and women, different sex hormones may preferentially select microbial species, and these species could carry out physiological functions, such as altering hormone levels, interacting with regional adiposity.

As presented in Table 1, there were 16 (16.7%) men and 19 (16.4%) women having metabolic syndrome according to the definition from the International Diabetes Federation[24]. Metabolic syndrome was deemed to associate with both regional adiposity and gut microbiota[25–28], therefore we examined the association between metabolic syndrome and regional fat mass in the studied population. The results (see Supplementary Table 1) indicated that after adjusted for age, education, and overall obesity, android fat mass was significant in men, not in women, whereas the gynoid fat mass was not significantly associated with metabolic syndrome in both sexes. Model further adjusted for the interaction between android and gynoid fat mass, the android fat mass was significantly associated with metabolic syndrome in both men and women. As suggested in the literature, Bacteroidaceae and Ruminococcaceae were often negatively associated with metabolic syndrome[27,28], our study could potentially underestimate the positive associations between android fat ratio and bacterial genera from these two families, and, conversely, overestimate the negative associations.

Comparing the sociodemographic characteristics between men and women, the patterns of education were different in the studied population. Education in men tended to cluster around middle school level, whereas in women it tended to aggregate at the two ends: illiterate and high school above. At the lifestyle behavior level, men and women tend to have different preference for food and different physical activity patterns[29–31]. Such differences in diet and physical activities could be impacted by one's education level and socioeconomic status[32,33]. Human studies showed that microbiome can also be affected by diet[34–36]; evidence from the animal models proposed that microbial richness and diversity could potentially be improved by physical exercises[31,37]. Thus, socioeconomic status impacted lifestyle behaviors could be another contributor to the observed microbial difference between men and women.

This study presents a high-resolution association between microbiome, android and gynoid fat ratio using precise measurement of fat distribution. The sex-induced difference in regional adiposity could potentially lead to the difference in the

species of functioning microbiome that were in turn modulating fat distribution. However, this study cannot provide the biochemical mechanism behind the observed difference. Follow-up in vitro or animal models are needed to establish the mechanism pathways. Moreover, among the 116 female participants, 59 of them self-reported absence of menstruation due to menopause, 51 reported normal menstruation, the other six had irregular menstruation due to other causes. Although, potentially due to the small sample size, the association between fat distribution and menopausal status was not detected in the female participants (see Supplementary Tables 2 and 3), we plan to address the menopausal status and directly measure blood hormone levels in a larger female sample size in our follow-up study. This will also guarantee capturing a larger variation of fat distribution in women. Further, although the gold standard for characterizing fat distribution are measurements from DXA and certain DXA devices have embedded algorithms to estimate visceral and subcutaneous fat masses, the algorithms for this estimation is not Asian-specific. Literatures suggest that android fat mass is more closely associated with metabolic syndrome in Asian populations[26], whereas visceral fat mass is more correlated with type 2 diabetes and other cardiovascular disease outcomes[38]. Therefore, we are planning to develop a visceral fat predictive model for Chinese population within our cohort, then we will be able to further our current finding by incorporating Asian-specific visceral and subcutaneous fat distribution information into our future studies. Finally, our study was conducted in a southern Chinese population, to our knowledge, there are no similar studies have been done in other ethnic groups. Thus, our current results potentially face limitations in generalizability. The WELL-China project has parallel sites in California, Taiwan, and Singapore[39]. This allows us to conduct comparative studies across myriad populations and ethnic groups in future.

In conclusion, our study proved the hypothesis of sex-specific association between gut microbiome and body fat distribution, demonstrated the need for further investigation to deepen the understanding of its mechanism. This is a valuable discovery for more accurate microbiome-based cardiovascular and metabolic outcomes prediction and treatment in future.

## Methods

**Samples and study setting**. The 222 participants were part of the Wellness Living Laboratory (WELL)-China project. Using cluster sampling, the 222 participants were chosen from the all the communities within one subdistrict from the WELL-China Site. WELL-China is a population-based longitudinal cohort located in the city of Hangzhou. The major objectives of WELL-China are to investigate upstream risk factors of chronic diseases, determinants of wellbeing, and develop approaches leading to precision health. The participants of WELL-China are recruited from 550,000 permanent residents of 12 sub-districts aged 18–80 years old. A combination of randomized sampling and quota sampling are applied. Baseline measurements in 2016 included anthropometric measurements and physical examinations, and in-person survey. Stool, fasting venous blood, hair, and nail samples were also collected from each participant. Stool and blood samples were stored immediately after collection at −80 °C. Informed consent forms were obtained from all participants in this study. The study has obtained the Institution Review Board approvals from both Stanford University (IRB-35020) and Zhejiang University (No. ZGL201507-3).

**Body composition and diet measures**. At baseline assessment, body composition data were obtained using DXA scan (Software version 11.40.004, GE Lunar Prodigy; GE Healthcare, Milwaukie, WI, USA), which measures fat mass, lean mass, bone mass, and bone density. The android region of interest (ROI) is defined as the area between the lower boundary at the pelvis cut and the upper boundary above the pelvis cut by 20% of the distance between the pelvis and neck cuts. The gynoid ROI is defined as the area between the upper boundary below the pelvis cut line by 1.5 times the height of the android ROI and the lower boundary at the two times the height of the android ROI below the upper boundary[40]. This study included three major fat distribution variables: android fat mass, gynoid fat mass, and total fat mass, which were used to derive android fat ratio and gynoid fat ratio. The former was defined as the ratio of android

fat mass to the total fat mass; similarly, the latter was the ratio of gynoid fat mass to the total fat mass. Android fat ratio has very weak correlation with total body fat, whereas gynoid fat ratio has intermediate correlation with total body fat. (see Supplementary Table 4) Diet measurements included carbohydrates, fat, protein intakes and total daily energy intakes were measured via a 26-item Food Frequency Questionnaire (FFQ) developed and validated within the same population[41]. Information on smoking and alcohol consumption was obtained from an in-person survey at baseline.

**Sequence processing and OTU mapping**. The detailed DNA extraction and 16S rRNA V4 region sequencing method is described in the Supplementary Method. To prepare the raw sequences for mapping and correct amplicon errors, the study employs the Divisive Amplicon Denoising Algorithm (DADA) from the open-source R package—DADA2. The implementation includes the following steps: filtering, trimming, dereplication, merging paired reads, chimera recognition and removal, and taxonomy assignment. This algorithm has been benchmarked and compared to four other popular algorithms: UPARSE, MED, Popular Mothur, and QIIME. DADA2 demonstrated the highest precision rate[42]. The denoised sequences are mapped to the GreenGenes reference database[43]. To ensure high accuracy, taxonomy is assigned at 97% identity. All the mapping and the following analysis were done in R (version 3.5.1). The plots for sequencing quality and error rate are included in the Supplementary Figs. 1 and 2.

**Global association of microbiome and fat distribution**. Based on our central hypothesis, the study visualized the unadjusted association between microbiome abundance and android fat ratio, and gynoid fat ratio, stratified by gender. We first transformed the microbial abundance at the species level using variance–stabilization–transformation from the DESeq2 package. Then, calculate the total microbiome abundance, and the total number of different microbiome species in each subject. We also examined the association between the microbiome abundance and diversity, the correlation coefficient was 0.99 ($p < 0.001$) in both men and women. Then divided the subjects into tertiles using the microbiome abundance, then diversity, and compared the two sets of tertiles. The two tertiles shared high level of agreements. Thus, the study employed the microbiome abundance tertile to represent both the abundance and diversity in each subject. The quartile cutoffs of android fat ratios for men and women were calculated separately. We generated four heat maps to visualize the unadjusted associations and employed Fisher's exact test and univariate linear regression to examine the independence between microbiome tertiles and fat distribution. When a potential quadratic relationship was observed in the heat map, the linear model included a quadratic term of the corresponding variable.

**Data preparation for testing**. To prepare the sequencing data for taxa level testing, a size factor for each sample was estimated using the median ratio method[44]. This step ensured samples at different sequencing depths were comparable. Each taxa's gene-wise dispersion was also estimated using maximum likelihood, modeling read counts as a negative binomial distribution. A gene-specific dispersion parameter was applied to adjust dispersions across gender stratified samples.

**Taxa level testing for associations**. Wald tests with taxa abundance as the primary outcome were performed using DESeq2 package[45]. Taxa were filtered in men and women separately, keeping only those with more than five raw counts in more than 7% of the samples. The filtering results in 324 taxa for men and 336 women. The remaining taxa were individually tested against android fat ratio and gynoid fat ratio as the main exposures in the final model. Given the importance of sex-difference to this study, the potential for sex-specific relationships with adjustment covariates, and the sample size this study fit separate models for men and women. Android and gynoid fat ratio were standardized. We included following covariates to the generalized linear model: age, daily fat intake, daily carbohydrate intake, daily total energy intake, smoking, alcohol consumption, body mass index (BMI), past-3-month antibiotic use, and sequencing batch. BMI was included in the model, as previous literatures have indicated that BMI, as a measurement of overall obesity, was associated with human gut microbiome[13,46,47]. Therefore the model will allow us to obtain the effects of fat distribution on the gut microbiome independent from the overall adiposity. There are potentially other methods to adjust for overall obesity, such as including total fat mass and height into the model instead of BMI. The total fat mass and height are associated with BMI in both men and women and explained 87% and 88% of the variability in BMI respectively (see Supplementary Tables 5 and 6). Therefore, the model still included BMI as the measurement of overall obesity. Effects were measured in Log2 fold change indicating the changes in the taxa abundance in terms of the power of 2. $p$-Values from the Wald tests for the coefficients of android and gynoid fat ratio were adjusted using Benjamini–Hochberg procedure[48]. We have also conducted sensitivity analysis to examine the effect without including the antibiotic use. (See Supplementary Table 7).

**Reporting summary**. Further information on research design is available in the Nature Research Reporting Summary linked to this article.

## Data availability

The sequencing data that support the findings of this study has been made publicly available at the NIH National Center for Biotechnology Information Sequence Read Archive (SRA) with BioProject ID PRJNA533934. Accession codes are provided in the supplementary material. The sample data are available from the corresponding author upon reasonable request.

## Code availability

The R programming codes related to the statistical analysis are publicly available as part of the supplementary materials.

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

## Acknowledgements

We would like to thank Susan Holmes from Stanford Department of Statistics for statistical consultation; the Nutrilite Health Institute Wellness Fund, the Cyrus Tang Foundation, and the Zhejiang University Education Foundation for funding.

## Author contributions

Y.M., X.M. and S.Z. designed the study protocol. Y.M. and K.S. led the statistical modeling and bioinformatics analysis. S.Z. and M.B. supervised the all the process of the projects and provided constructive analytical suggestions. X.M. supervised the sample sequencing and data quality. Y.R. and L.C. provided critical comments during the manuscript revision.

## Additional information

**Competing interests:** The authors declare no competing interests.

