## [Peer Review File · Nature Communications]

Reviewers' comments:

Reviewer #1 (Remarks to the Author):

Min et al have examined the statistical association of body fat distribution (BFD) and the composition of the gut microbiome in men and women. This is an important question and they appropriately address the question of sex differences in this context. Their major claim is that BFD has different effects on the gut microbiome in each sex. They used DXA to define BFD as the android (upper body) fat ratio or the gynoid (lower body) fat ratio. Though never clearly stated, they appear to be taking the ratio each the mass of android or gynoid fat to total body fat. Their first analysis of the association of microbiome tertiles unadjusted for potential confounders and stratified by sex. The full model of Taxa adjusts for age and BMI and diet (fat and carbohydrate intake) and alcohol intake. They found sex specific signatures underlie relationships to android ratio in men and women. No metabolic data are provided, and android fat distribution is strongly associated with metabolic health (insulin resistance, lipids, BP, etc) and sex has partly independent effects on metabolic outcomes. Thus, the associations of android vs gynoid fat with the microbiome may merely reflect metabolic status. It is impractical to use DXA to assess regional fat distribution, so the paper would have a greater impact on the field if models were created to address the influence of waist and hip circumference (or their ratio) were also included.

There are a number of question/concerns about the approach and conclusions that were drawn:

1. The ratios of each region of fat to total fat to assess fat distribution is reasonable - as there is no fully accepted method to define BFD, but it is not without potential problems that should have been discussed. For example, the android ratio is likely correlated with total body fat and the shape of the relationship may be different in each sex. This important question was not directly addressed -- specifically in the population they studied.
2. Obesity – not phenotyped with respect to body fat distribution – has documented effects on the microbiome. Thus, an important question for the authors to address whether the sex-specific effects of BFD on the microbiome they have observed are independent of overall adiposity (% body fat). Some of their models adjust for BMI (a proxy for body fat/level of obesity), but the question of whether adjustment for BMI alone attenuates associations of fat ratios and microbiome readouts is not specifically discussed. Furthermore, the rationale for using BMI is not clear. There are other approaches, none perfect, to assess the influence of the mass of each depot, total fat and perhaps height and age on the microbiome in each sex.
3. Please explicitly provide the equations used to calculate Android and Gynoid ratio. The methods section implies that $AR = AF/TF$ and $GR = GF/TF$ and in this case $AR+GR$ should be close to 80-90% since AF and GF account for the majority of TF. However, in Table 1 AR is approx. 10% and GR 17%. It is possible that Android Ratio was defined as Android Fat/Total Android Mass and Gynoid Ratio similarly (this is sometimes used in reports of DXA results) but the interpretation of such ratios is unclear.
4. Assessing the association of BFD and the microbiome is difficult because android (A) fat mass is strongly positively correlated to metabolic syndrome characteristics while gynoid (G) fat has a negative (protective effect) in both sexes. Arguably, a model should include the effect of A, G and the interaction of A and G fat. This question is important because obesity per se likely influences the microbiome, and to this reviewer's knowledge, it is not known if the relationships among these variables are the same in men and women. In any event, these basic questions should have been better addressed and discussed in this manuscript.
The changes in the microbiome could reflect metabolic status, independent of total body fat and possibly fat distribution. It is important for the field to have this to have issue to be addressed -- it at least should be discussed.
5. DXA does not provide an estimate of visceral fat, which has a strong effect on metabolic health and possibly the microbiome. Thus, 'android' fat includes abdominal subcutaneous and visceral. Many sex differences in metabolism are attenuated or eliminated after adjustment for visceral fat. This issue should be mentioned as a limitation.

Specific points:

Title: "sex confounds....".... to a biologist this is an odd description of what is, perhaps, a sex difference in the microbiome that is dependent on BFD.

Sex is used here as a biological variable, so the term 'gender' is not correct.

Model: The influence of total fat (or % body fat) vs fat distribution should be considered.

Sex steroids vs sex chromosomes. The model included 'sex steroids' but does not address the potential, independent role of the sex chromosomes (e.g. see Reue review PMID 28284880).

The introduction refers to studies in mice without mention of their sex.

The influence of menopausal status in the women should be addressed (as the population is on average 50 years with a SD of 14) and this factor is known to affect fat distribution.

The population is poorly described. What criteria were used to select the 222 participants out of the total WELL study? Please provide medical history/medication information (should be available) with mean age 51y comorbidities are likely and may influence study outcomes. If no circulating values of glucose, lipids, etc are available this should be acknowledged as a study limitation.

The authors imply that sex hormones, e.g. estrogen in women that affect fat distribution may also affect the microbiome. However, it is unclear whether there is a major difference in fat distribution between men and women in this study. The P value is not provided but the mean WHR of 0.89 in women is in the "apple" (android) shape range.

L76-115, Global association by Sex. The authors used microbiome abundance and fat distribution as categorical variables here and failed to detect statistically significant associations. Using continuous variables, i.e. WHR instead of quartiles etc, may be a better option.

Methods, L297: The most recent R version is 3.5.1 so it is unlikely that the authors used v15.6.0. Figure 3 is duplicated and likely redundant.

Reviewer #2 (Remarks to the Author):

Min and colleagues address a thoughtful question in their manuscript about whether the association between gut microbiome and fat distribution differs by sex in Chinese adults. The paper is well written and contributes to the field. I have a few comments for improvement and clarification of the manuscript.

Since the main findings are based on sex-specific association between gut microbiome and fat distribution, sex could be an effect modifier rather than confounder.

The authors evaluated android fat ratio and gynoid fat ratio, separately in relation to gut microbiome, but it would be interesting to look at the combined effect of android fat and gynoid fat (e.g. the ratio of android to gynoid fat) because this ratio may reflect cardiometabolic risk better than individual fat variables and the distribution of the ratio of android to gynoid fat looks different between men and women.

Total energy intake should be included as a covariate in the multivariable-adjusted model.

Otherwise, fat intake and carbohydrate intake should be included in the model after adjusting for energy intake.

Since obesity is a main outcome, lack of information about physical activity may be a limitation in interpreting the results.

Android fat and gynoid fat need to be defined with detailed information of measurement (e.g. the specific location and level of body).

Reviewer #3 (Remarks to the Author):

Increasing evidence suggests an important role for the gut microbiome, in the development of obesity and weight gain. There is a distinct pattern in which body fat is distributed in men and women throughout the body. A role for the regulation by sex hormones has been indicated, in fat distribution. Men gain weight around their abdomen (android pattern), whereas women gain weight on their hips (gynoid pattern). According to the authors, gender differences impact microbiota composition which may in turn contribute to abdominal obesity, and distinct fat distribution patterns. Furthermore, they hypothesize that men may be more susceptible than women, for abdominal adiposity and related health outcomes. Knowledge gained on how microbiota contribute to these distinct patterns of adiposity may serve as therapeutic avenue for microbial intervention strategies. In the current manuscript, Min et al explore this angle by looking at microbiota composition in men and women from an existing Chinese cohort and drawing the correlations between microbiota abundance with fat distribution and BMI. They provide some interesting data on microbiota composition among men and women and differences in non-overlapping strains that are associated positively and negatively with android and gynoid distribution. They infer that sex hormones and microbiota may bi-directionally influence adiposity in men and women and make a case for the need for special attention to gender differences in microbiome-obesity studies.

One of the main concerns is that the authors have not paid attention to literature on the impact of sex differences in microbiota composition. There have been a couple of recent papers, exploring the relation between the microbiota, gender differences and adiposity. Specifically, the paper from Haro et al (PLoS One. 2016 May 26; 11(5):e0154090. Doi), explores the influence of gender differences and BMI on microbiota composition; and draws very similar conclusion as the current manuscript. The nature of how the results were presented in the manuscript did not allow for direct comparison of the data with the published article. This reviewer suggests that the current manuscript needs to be revised, taking these observations and inferences into consideration.

There is another paper (Appl Environ Microbiol. 2006 Feb; 72(2): 1027-33) that explored the impact of demographic factors such as gender, age and country of residence (Europe) on microbiota composition. It may be interesting to compare the data from this study with the observations from the earlier paper, to see how this data differs from the European cohorts (in terms of broader distribution among men and women).

Additional aspects that need to be addressed:

- 1) Since this study was conducted only in Chinese cohorts; generalizability to other ethnic cohorts and other countries is somewhat limited (as their dietary, cultural and lifestyle habits and genetic susceptibility can vary dramatically, which may contribute to differential microbial abundance). Adding a brief explanation on the limitation of the current study or providing a detailed justification to their conclusion is helpful.
- 2) It is not clear how many of the women subjects are in menopause, which may confound the interpretation, due their altered hormonal status.
- 3) The participating subjects have an average BMI of 23. That means they are still within the normal range of body weight. It is not clear if the fat distribution is large enough to be measurable and classified into android and gynoid patterns.

- 4) It was not clear, if microbiota composition can be influenced by Fat Free Mass (FFM). Related to this idea; an Australian paper (J Pediatr Gastroenterol Nutr. 2018 Jan;66(1):147-151. Doi) explores the relation between microbiota composition and fat-free mass, particularly in male children.
- 5) Host physiology and sex hormones may independently exert influence on fat distribution, which may be much larger than the contribution by microbiota.
- 6) 16S rRNA sequencing used for microbial compositions is not deep enough for achieving better resolution of microbiota at strain level and to capture the low abundant species. However, authors are indeed familiar with this limitation.
- 7) Line 137: while describing the results from the Table 2: about positive association of microbial strains among males – It is not clear why they do not include Eggerthella from Coriobacteriaceae family with effect size more than 8, in their results. Also, it not clear why another strain Gemminger from Ruminococcaceae family has effect size of 7.7, which is closer to the largest value (8.2), not included.
- 8) Line 223: Typo- 'one' the other hand. It should be "on" the other hand
- 9) Line 238: Through illustration in the figure 3; authors conclude that the interplay between sex, microbiome collectively impact fat distribution. However, it is not clear if fat distribution per se impacts differential microbial populations or the differences in microbiota contribute to the differential fat distribution. Also, there is no discussion (or speculation) on how sex hormones contribute to differences in microbiota populations and further impacting the fat distribution.
- 10) Educational status of an individual may impact their knowledge of personal hygiene and achievement of socio-economics status that generally contribute to their lifestyle choices. However, there is no discussion on how the information on educational status has been factored into the current findings.

#	REFREEE 1 Comment	Response	Action	Corresponding Line Number
1	The ratios of each region of fat to total fat to assess fat distribution is reasonable - as there is no fully accepted method to define BFD, but it is not without potential problems that should have been discussed. For example, the android ratio is likely correlated with total body fat and the shape of the relationship may be different in each sex. This important question was not directly addressed -- specifically in the population they studied.	The referee raises a good point. Thus, we further explored the correlation of the android fat ratio and the total body fat by conducting a correlation test of the android fat ratio and total fat mass and plotting out the shape of the relationship in both sexes as well. The correlation coefficients in men and women are 0.05 and 0.20 with p-values 0.636 and 0.032 respectively.	Added this point to the method section under the "Body Composition and Diet Measures" subtitle; added the plots of the relationship and the result from correlation tests to the supplementary document (Appendix 1).	335
2	Obesity – not phenotyped with respect to body fat distribution – has documented effects on the microbiome. Thus, an important question for the authors to address whether the sex-specific effects of BFD on the microbiome they have observed are independent of overall adiposity (% body fat). Some of their models adjust for BMI (a proxy for body fat/level of obesity), but the question of whether adjustment for BMI alone attenuates associations of fat ratios and microbiome readouts is not specifically discussed. Furthermore, the rationale for using BMI is not clear. There are other approaches, none perfect, to assess the influence of the mass of each depot, total fat and perhaps height and age on the microbiome in each sex.	This is an interesting point. To address the referee's comment, we examined the linear relationship between BMI, height and total fat mass. There are positive associations between BMI and total fat mass, and negative associations between BMI and height with both men and women. Together, height and total fat mass explain 87% of the variance of BMI in men and 88% of the variance in women. Thus, we presume the results could potentially be similar. We would love to further compare this way of adjusting obesity with using BMI using a more sufficient sample size in our follow up study. We also examined the correlation between BMI and android fat ratio, which turned out to be 0.07 in men (p = 0.500) and 0.35 in women (p < 0.005). Then we examined the correlation between BMI and gynoid fat ratio, obtained a correlation coefficient of -0.66 in men (p < 0.005) and -0.57 in women (p < 0.005). The primary reason we used BMI in the model is to stay consistent with the literature that evaluating the effects that obesity has on microbiome, thus obtain the independent effect that fat distribution has on microbiome. When adjusting for BMI alone in the model, it would be less likely to decrease the coefficients of android fat ratio and may decrease the coefficients of gynoid fat ratio.	Added the rationale of using BMI in this study in the method section, under "Taxa Level Testing for Associations Section". Added results of the linear model testing associations between BMI and total fat mass and height. Added the results of the correlation tests of BMI and android fat ratio and gynoid fat ratio in men and women to the supplementary document (Appendix 2 and 3).	382
3	Please explicitly provide the equations use to calculate Android and Gynoid ratio. The methods section implies that AR = AF/TF and GR = GF/TF and in this case AR+GR should be close to 80-90% since AF and GF account for the majority of TF. However, in Table 1 AR is approx. 10% and GR 17%. It is possible that Android Ratio was defined as Android Fat/Total Android Mass and Gynoid Ratio similarly (this is sometimes used in reports of DXA results) but the interpretation of such ratios is unclear.	It is a good suggestion to include the explicit definition of android fat ratio and gynoid fat ratio, which are the ratio of android fat mass to the total fat mass and the ratio of the gynoid fat mass to the total fat mass. DXA defines between the lower boundary at the pelvis cut and the upper boundary above the pelvis cut by 20% of the distance between the pelvis and neck cuts. Android fat is a subset of trunk fat, which includes all the fat in the region between neck line and upper pelvis. Trunk fat mass + gynoid fat mass on average accounts for 76% of the total fat mass in our studied population. AR + GR on average accounts for 27.9% of the total fat mass.	We included the explicit definitions of android fat ratio and gynoid fat ratio in the method section under the subtitle "Body Composition and Diet Measures"	333
4	Assessing the association of BFD and the microbiome is difficult because android (A) fat mass is strongly positively correlated to metabolic syndrome characteristics while gynoid (G) fat has a negative (protective effect) in both sexes. Arguably, a model should include the effect of A, G and the interaction of A and G fat. This question is important because obesity per se likely influences the microbiome, and to this reviewer's knowledge, it is not known if the relationships among these variables are the same in men and women. In any event, these basic	This is a great suggestion to address the relationship of metabolic syndrome and regional fat mass in men and women, as it is not known whether different sexes have the same relationship. To address this interesting point, we constructed 4 models in each sex to examine the associations in men and women. All four models use metabolic syndrome as the outcome, model 1 only included android fat mass adjusted for age, education, BMI; model 2 only included gynoid fat mass adjusted for the same covariates; model 3 included both android and	We included the results of the 4 newly constructed models of men and women in the Appendix 4, added this point to the discussion section under the subtitle "Effects from Fat Distribution and Metabolic Syndrome".	256

questions should have been better addressed and discussed in this manuscript. The changes in the microbiome could reflect metabolic status, independent of total body fat and possibly fat distribution. It is important for the field to have this to have issue to be addressed -- it at least should be discussed.	gynoid fat mass adjusted for the same covariates; model 4 included android, gynoid fat mass and their interaction adjusted the same covariates. We would like to include the insights from these models into our discussions.	
DXA does not provide an estimate of visceral fat, which has a strong effect on metabolic health and possibly the microbiome. Thus, 'android' fat includes abdominal subcutaneous and visceral. Many sex differences in metabolism are attenuated or eliminated after adjustment for visceral fat. This issue should be mentioned as a limitation.	The referee raised a valid concern, which we also think can potentially attenuate the signal that we are trying to measure here. Thus, we would like to expand this point on top of our original wordings included in the discussion section.	292
Specific Points		
1 Title: "sex confounds...." ... to a biologist this is an odd description of what is, perhaps, a sex difference in the microbiome that is dependent on BFD.	To avoid this issue, we will modify the title.	1
2 Model: The influence of total fat (or % body fat) vs fat distribution should be considered.	We understand the referee's point here. Indeed, total fat (% of body fat) and its association with gut microbiome is an interesting topic. In the introduction, we also cited studies that are exploring this topic. In our model, we included BMI, which is widely recognized as an indicator of overall obesity, on top of that, we included fat distribution measurements to tease out the association between fat distribution and gut microbiome independent from the overall obesity.	379
3 Sex steroids vs sex chromosomes. The model included 'sex steroids' but does not address the potential, independent role of the sex chromosomes (e.g. see Reue review PMID 28284880).	Thank you for pointing out this interesting article. As the referee mentioned, this article mainly addressed the effect of x chromosome dosage on food intake, and further on adiposity. In our study, we are currently focusing on the association between microbiome and fat distribution in men and women. This article didn't expand regarding the effect of x chromosome has on fat distribution. In our future studies examining adiposity related effects, we will include the theories proposed in this recommended article.	-
4 The introduction refers to studies in mice without mention of their sex.	One of the studies didn't mention litter sex in one study, the other one only included male mice in the study. Citing these two articles mean to show the associations between microbiome and obesity in experiment animals. All the human studies are having sex mentioned.	50
5 The influence of menopausal status in the women should be addressed (as the population is on average 50 years with a SD of 14) and this factor is known to affect fat distribution.	The referee raised a great point. Among all the 116 female participants that are included in the final analysis, 59 of them self-reported having menopause. To address referee's comments, we constructed 5 statistical models to examine the association between menopause and overall obesity, menopause and android fat ratio, and menopause and gynoid fat ratio. We included age and education level as the covariates for the above 5 models. We also compared the android fat ratio between men and post menopause women adjusted for age, education, and BMI. We didn't observe any significant associations regarding the three associations among women, we observed significant difference in android fat ratio between men and post-menopause women. We would like to add related literatures to our discussion section, as this is truly an important point.	286

6	The population is poorly described. What criteria were used to select the 222 participants out of the total WELL study? Please provide medical history/medication information (should be available) with mean age 51y comorbidities are likely and may influence study outcomes. If no circulating values of glucose, lipids, etc are available this should be acknowledged as a study limitation. The authors imply that sex hormones, e.g. estrogen in women that affect fat distribution may also affect the microbiome. However, it is unclear whether there is a major difference in fat distribution between men and women in this study. The P value is not provided but the mean WHR of 0.89 in women is in the "apple" (android) shape range.	The referee has several good points here. We would like to report disease history and more biomarker information in the article.	Added description of the sampling of the studied population in the method section under "Samples and Study Setting" subtitle. Included more detailed disease histories and biomarkers in the Table 1, also applied statistical tests to compare male and female samples and provided p-values.	84, 311
7	The authors imply that sex hormones, e.g. estrogen in women that affect fat distribution may also affect the microbiome. However, it is unclear whether there is a major difference in fat distribution between men and women in this study. The P value is not provided but the mean WHR of 0.89 in women is in the "apple" (android) shape range.	The referee raised a good point. As addressed in the previous response, we have conducted t test to examine the difference between the male and female WHR, and the result was significant at $p = 0.005$.	Added p-values for all the characteristics are included in Table 1.	84
8	L76-115, Global association by Sex. The authors used microbiome abundance and fat distribution as categorical variables here and failed to detect statistically significant associations. Using continuous variables, i.e. WHR instead of quartiles etc, may be a better option.	Here we meant to stay consistent with the following tests conducted in the research. Moreover, as regional fat ratios are significantly correlated with the WHR in both men and women, we still would like to stay with the regional fat ratios as they are precise measurements in this case and consistent with the follow-up analysis.	We included the results of the correlation tests in the supplementary Appendix 4.	-
9	Methods, L297: The most recent R version is 3.5.1 so it is unlikely that the authors used v15.6.0.	Thanks for pointing it out.	Corrected in the paper, method section	348
10	Figure 3 is duplicated and likely redundant.	Thanks for your recommendation, to avoid redundancy, we will remove the figure and the relevant wording indicating the figure.	Figure 3 and relevant wording removed.	-

#	Comment	Response	Action	Corresponding Line Number
1	Since the main findings are based on sex-specific association between gut microbiome and fat distribution, sex could be an effect modifier rather than confounder.	To avoid this confusion, we would like to change the title using non-causal inference related terms.	We proposed a new title as "Sex-Specific Association Between Gut Microbiome and Fat Distribution".	1
2	The authors evaluated android fat ratio and gynoid fat ratio, separately in relation to gut microbiome, but it would be interesting to look at the combined effect of android fat and gynoid fat (e.g. the ratio of android to gynoid fat) because this ratio may reflect cardiometabolic risk better than individual fat variables and the distribution of the ratio of android to gynoid fat looks different between men and women.	The referee raises an interesting point. Literature suggested that waist-to-hip ratio could be a better predictor of cardiovascular outcomes. Currently we are focusing on the android fat ratio and gynoid fat ratio separately, as we are still at the early stage to tease out the sex modified relationship between microbiome and regional adiposity. Taking a ratio of android and gynoid fat could lose the regionality in the measurement. As it is indeed a useful predictor of cardiovascular disease, we would like to include this compound measure in our future study examining anthropometrics and cardiovascular outcomes.	No change has been made regarding this point.	-
3	Total energy intake should be included as a covariate in the multivariable-adjusted model. Otherwise, fat intake and carbohydrate intake should be included in the model after adjusting for energy intake.	The referee has a good point here. As in our original model, we simply adjusted for daily fat intake and daily carbohydrate intake. We should also adjust for total energy intake.	Added total energy intake to the model. The results turned out to be a subset of the original results. We updated table 2 and 3, figure 2 accordingly. And updated the wording in the result and discussion sections.	140, 147, 377
4	Since obesity is a main outcome, lack of information about physical activity may be a limitation in interpreting the results.	The referee has a good point regarding the association between physical activity and obesity. Physical activity may potentially impact fat distribution as well. There is a bit of controversy regarding how physical activity impacts the gut microbiota. This is an interesting point, and we would like to add this and relevant citations into our discussion section.	Expanded the lifestyle behavior sentence by including the discussions on the physical activity in the discussion section under the potential explanation subtitle.	270
5	Android fat and gynoid fat need to be defined with detailed information of measurement (e.g. the specific location and level of body).	Thank you for pointing it out. Yes, we would like to explicitly define the two definitions.	We extracted the definition from the GE Lunar DXA user manual and added to the method section, body composition measurement paragraph.	326

#	Comment	Response	Action	Corresponding Line Number
1	One of the main concerns is that the authors have not paid attention to literature on the impact of sex differences in microbiota composition. There have been a couple of recent papers, exploring the relation between the microbiota, gender differences and adiposity. Specifically, the paper from Haro et al (PLoS One, 2016 May 26;11(5):e0154090. Doi), explores the influence of gender differences and BMI on microbiota composition; and draws very similar conclusion as the current manuscript. The nature of how the results were presented in the manuscript did not allow for direct comparison of the data with the published article. There is another paper (Appl Environ Microbiol. 2006 Feb;72(2):1027-33) that explored the impact of demographic factors such as gender, age and country of residence (Europe) on microbiota composition. It may be interesting to compare the data from this study with the observations from the earlier paper, to see how this data differs from the European cohorts (in terms of broader distribution among men and women).	Many thanks to the referee for sharing these two articles with our team. The two articles addressed an interesting issue regarding gender difference in microbiome in relationship to BMI (a measurement of overall obesity). We consider these two studies are predecessors to our study, as we took one step further considering the effect of fat distribution independent from the overall obesity level. We think these two studies are definitely worth citing and discussing.	Added this article into the introduction section and added paragraphs of discussion in the discussion section.	54 ,21
2		The referee raises a good point. We would like to further expand our comparison section by including the results from this recommended article.	Added the comparison in the discussion section, under the microbiome, fat distribution and sex subtitle.	217
Additional Points				
1	Since this study was conducted only in Chinese cohorts; generalizability to other ethnic cohorts and other countries is somewhat limited (as their dietary, cultural and lifestyle habits and genetic susceptibility can vary dramatically, which may contribute to differential microbial abundance). Adding a brief explanation on the limitation of the current study or providing a detailed justification to their conclusion is helpful.	It is a good point that in the end of the paper we shall address the limitation in generalizability to other ethnic groups.	Added this point to the limitation section of the paper in the end, proposed the next step.	300
2	It is not clear how many of the women subjects are in menopause, which may confound the interpretation, due their altered hormonal status.	The referee raised a great point. Among all the 116 female participants that are included in the final analysis, 59 of them self-reported having menopause. To address referee's comments, we constructed 5 statistical models to examining the association between menopause and overall obesity, menopause and android fat ratio, and menopause and gynoid fat ratio. We included age and education level as the covariates for the above 5 models. We also compared the android fat ratio between men and post menopause women adjusted for age, education, and BMI. We didn't observe any significant associations regarding the three associations among women, we observed significant difference in android fat ratio between men and post-menopause women. We would like to add related literatures to our discussion section, as this is truly an important point.	We included the test results in the supplementary material (see Appendix 6), addressed the menopause status in the discussion under the limitation future direction subtitle.	286
3	The participating subjects have an average BMI of 23. That means they are still within the normal range of body weight. It is not clear if the fat distribution is large enough to be measurable and classified into android and gynoid patterns.	The referee has a valid concern here. We added some analysis regarding the association between metabolic syndrome and fat distribution in our studied population. We were above to identify android fat mass was positively associated with metabolic syndrome after adjusted for age, education, and BMI. This could provide an indirect evidence showing the effect of fat distribution is measurable in our studied population.	Added the analysis results in the supplementary Appendix 5.	-
4	It was not clear, if microbiota composition can be influenced by Fat Free Mass (FFM). Related to this idea; an Australian paper (J Pediatr Gastroenterol Nutr. 2018 Jan;66(1):147-151. Doi) explores the relation between microbiota composition and fat-free mass, particularly in male children.	The referee provided an interesting article, which examined the association between the fat free mass and fecal microbiome among 2-3-year-old Australian boys. Currently we only have adults included in our study. We would like to explore the potential effect fat free mass may introduce to the interplay among fat distribution, sex, and microbiome in our further studies.	No change has been made regarding this point.	-
5	Host physiology and sex hormones may independently exert influence on fat distribution, which may be much larger than the contribution by microbiota.	The referee brought up a good point. We try not to compete the effect on fat distribution from sex hormones with the effect from microbiota. As potentially, there could be a complex feedback	No change has been made regarding this point.	-

		loop between the interaction within microbiota and regional adiposity, this topic is a bit beyond the scope of our current study. We should definitely include more in-depth discussion regarding this topic in our follow-up studies.		
6	16S rRNA sequencing used for microbial compositions is not deep enough for achieving better resolution of microbiota at strain level and to capture the low abundant species. However, authors are indeed familiar with this limitation.	Yes, we are aware of the limitation. We would consider metagenomic approach in our follow-up studies.	No change has been made regarding this point.	-
7	Line 137: while describing the results from the Table 2: about positive association of microbial strains among males – It is not clear why they do not include Eggerthella from Coriobacteriaceae family with effect size more than 8, in their results. Also, it not clear why another strain Gemminger from Ruminococcaceae family has effect size of 7.7, which is closer to the largest value (8.2), not included.	Thank you very much for pointing it out. That was originally a typo on our end. However, as recommended we added total energy intake into the model, the taxon from Eggerthella genus reduced in effect size, as its effect was modified by the total energy intake.	We updated the results in table 2 and 3, and figure 2 accordingly.	155
8	Line 223: Typo- 'one' the other hand. It should be "on" the other hand	There was indeed a typo.	Typo has been corrected	214
9	Line 238: Through illustration in the figure 3; authors conclude that the interplay between sex, microbiome collectively impact fat distribution. However, it is not clear if fat distribution per se impacts differential microbial populations or the differences in microbiota contribute to the differential fat distribution. Also, there is no discussion (or speculation) on how sex hormones contribute to differences in microbiota populations and further impacting the fat distribution.	The referee raised two interesting points. As recommended, we removed figure 3 from the discussion section since it's duplicating the information under the subtitle sex hormone centered explanation. We would like to add further explanations regarding these two topics: 1) different sex hormones have been shown preferentially selecting microbial species, in turn, microbiota also alters the systematic hormone levels. 2) these preferentially selected microbial species could potentially have differential effect on regional adiposity depending on their functionalities of producing short-chain fatty acid, increasing lipoprotein lipase, and stimulating local angiogenesis. 3) Previous studies have shown obesity alters the gut microbiome, however there is a lack of literature exploring how does regional adiposity impact regional microbiota.	We expanded our original discussion to further address the two questions raised by our referee. The revised wording can be found in the discussion section, under the "sex hormone centered explanation" subtitle.	233, 244, 252
10	Educational status of an individual may impact their knowledge of personal hygiene and achievement of socio-economics status that generally contribute to their lifestyle choices. However, there is no discussion on how the information on educational status has been factored into the current findings.	The referee again raises an interesting point. We conducted a chi squared test of independence to examine the association between education levels and gender and discovered that men and women tend to have different education levels.	We described the potential impact of education on lifestyle behaviors, and added this point into our discussion section, under the "social demographic and lifestyle behavior effects" subtitle.	270

Reviewers' comments:

Reviewer #1 (Remarks to the Author):

Min et al. and made a very careful reply and added supplemental data that address the reviewers concerns. Their main point is that are sex-specific microbiome signatures that depend on fat distribution. However, it is still not clear to what extent this relationship is influenced by the menopausal status and total body fat, and whether the relationship depends on the co-variation with the metabolic syndrome. The authors now present associations of metabolic syndrome (M.S.) and body fat distribution, but they enter these factors into in their multivariate models. Thus, we cannot conclude if M.S. characteristics influence the relationship of body fat, fat distribution and microbiome diversity.

The abstract is not written in a way that is readily accessible/understandable to a general audience (expected from Nat. Comm) – the second to last sentence is especially unclear.

Although the authors added data to show, as expected, the correlation of BMI and total body fat is high, the rationale that many studies use BMI rather than body fat for adjustment in their models is not clear. They simply say that many other studies use. However, most studies adjust for BMI rather than body fat because they do not have a direct measure of the latter. It is especially problematic when comparing the sexes, as women higher % fat than men for the same BMI. Thus, it makes sense to use total body fat as co-variate in this study.

About half the population of women studied is post-menopausal so it is unclear why the analyses of association with microbiome signatures was not adjusted for menopause.

The average WHR for the women is 0.89, which makes them rather 'apple' shaped –not unexpected for an Asian population. Although this is still statistically lower than it men (now shown) as indicated in the reply, the biological significance is less clear and not addressed -- the apparently limited range of WHR /fat distribution in women may provide a less than optimally powered test of the hypothesis about body fat distribution as clearly 'pear shape' women (WHR < 0.75) are under-represented. This limitation is partially acknowledged but not explicitly made.

Reviewer #2 (Remarks to the Author):

Thank you for your response and clarification.

Reviewer #3 (Remarks to the Author):

[No further comments for authors.]

1. Reviewer 1: Min et al. and made a very careful reply and added supplemental data that address the reviewer's concerns. Their main point is that are sex-specific microbiome signatures that depend on fat distribution.

---Proposed response---

Below, you'll see we separated the three major concerns raised and offered responses accordingly. We really appreciate the time and efforts the reviewers and the editors put into this paper. The suggestions have clearly made our thinking sharper on these issues. Thank you very much.

2. Reviewer 1: The authors now present associations of metabolic syndrome (M.S.) and body fat distribution, but they enter these factors into their multivariate models. Thus, we cannot conclude if M.S. characteristics influence the relationship of body fat, fat distribution, and microbiome diversity.

---Proposed response---

Firstly, thank you very much for raising the question regarding the metabolic syndrome. It is truly a

complicated multi-factored condition. M.S., when mentioned in this study, was defined using the International Diabetes Federation criteria for Chinese population.¹

[Redacted]

(Shown in table 1) Waist

circumference equaling to or exceeding 90 cm in men and 80 cm in women is a mandatory criterion for diagnosing metabolic syndrome. To be eventually classified as metabolic syndrome, one also needs to fulfill at least two out of the four secondary indicators. Thus, all the diagnosed metabolic syndrome patients will have sex-wise some variability in the waist circumference though they are all above the cut-off points but have more variabilities in the manifestations of the remaining indicators. Regarding our current model, one of our main exposures is the android fat ratio, it is closely related to the waist circumference but further adjusted for total fat mass, the four remaining indicators were not explicitly adjusted in the current model. We think metabolic syndrome and its potential influence on microbiome abundance and diversity may overlap with our current findings to a certain extent as the main manifestation of metabolic syndrome is high waist circumference. The difference is our model tests the android fat ratio as a continuous variable, whereas in metabolic syndrome classification, the waist circumference is included as a binary variable. Further investigation of the association between the remaining indicator and microbiome will definitely be interesting and necessary to explore in future studies.

3. Reviewer 1: The abstract is not written in a way that is readily accessible/understandable to a general audience (expected from Nat. Comm) – the second to last sentence is especially unclear.

---Proposed response---

Thank you for suggesting that we rework the Abstract to make it more accessible/understandable. In particular, we paid attention to clarifying the second to last sentence. Please see:

"The gut microbiome has been linked to obesity. However, the sex-specific association between microbiome and fat distribution has been understudied. Here we demonstrated that even though both sexes had the same association between the fat distribution and the gut microbiome characteristics, including overall abundance and diversity, there were sex-specific microbiome signatures contributing to the above relationship. Comparing the taxa associated with the android fat ratio in men and women, there were no species-level overlaps. Although there were overlaps at the genus and family levels, such as *Holdemanella*, and *Gemmiger*, they actually had the opposite associations with fat distribution in men and women. Our findings suggested that sex-specific fat distribution may have sex-specific relationships with the composition of the microbiome. When studying the gut microbiome and abdominal obesity-related disease outcomes, the sex-specific microbiome signatures for fat distribution should not be overlooked."

4. Reviewer 1: Although the authors added data to show, as expected, the correlation of BMI and total body fat is high, the rationale that many studies use BMI rather than body fat for adjustment in their models is not clear. They simply say that many other studies use. However, most studies adjust for BMI rather than body fat because they do not have a direct measure of the latter. It is especially problematic when comparing the sexes, as women higher % fat than men for the same BMI. Thus, it makes sense to use total body fat as co-variate in this study.

---Proposed response---

We appreciate the reviewer noting that the unique metrics we have in our data (e.g., DXA derived fat distribution measures) which give us a unique set of choices about how best to address overall body fat. First, it's important to note that we chose to incorporate total fat as part of the "android fat ratio" which is $(\text{android fat})/(\text{total fat})$.

Regarding the Reviewer's point that: "It is especially problematic when comparing the sexes, as women higher % fat than men for the same BMI."

We agree that this is an important point - that is, that the coefficient for BMI could possibly be fit incorrectly and therefore confounded with sex. We had that same concern that if the BMI coefficient were to have been fit using both men and women then that BMI coefficient would have reflected some sort of weighted-average of men with big BMI and women with small BMI. That fitting procedure would have been problematic, and (importantly) would not have addressed the variation we wanted to adjust for (i.e., overall fat). To head off that concern, we fit two separate models - one for men and the other for women. Because we fit the models separately, the BMI coefficient for men reflects the correlations for just men, and the coefficient on BMI from the women's model similarly is for only women. Therefore, these coefficients (i.e., "nuisance parameters") do not suffer from the concern the Reviewer was right to point to. (This same argument holds for all of the covariates in our models.)

To help make the model-fitting procedure clearer for readers we modified the manuscript (page 9, lines 373-375) to say "*Given the importance of sex-difference to this study, the potential for sex-specific relationships with adjustment covariates, and the sample size this study fit separate models for men and women.*"

5. Reviewer 1: About half the population of women studied is post-menopausal so it is unclear why the analyses of association with microbiome signatures was not adjusted for menopause. The average WHR for the women is 0.89, which makes them rather ‘apple’ shaped – not unexpected for an Asian population.

---Proposed response---

First, this is a nice point and, like we discussed in our response to Reviewer 3's questions about menopause, we think that future studies should pursue this idea in more detail (see second part of this response below). In our response to Reviewer 3, we created several analyses to give some sense of the correlations - these can be found in Appendix 6. Like we discussed with Reviewer 3, our data set - while interesting - is underpowered to adequately investigate all possible subgroups. We agree that menopause is likely to be a big modifier; we hope that we can recruit more women into our study, so we can look into the subgroup differences.

-Second, it's important to note that the argument used by both Reviewer 3 and Reviewer 1 (that differences in fat-distribution and microbiome are being driven by women who are post-menopausal and premenopausal) is fundamentally an argument about changes in sex-hormones being related to fat-distribution and the microbiome. That is, this is consistent with the overall argument we are making. Further, we are quite excited by the Reviewers' ideas because the insights about menopause suggests that a within-woman (pre-/post-menopause) study design as a potentially very strong follow-up study to the current study (but it would require a repeated-measures data set which we do not currently have).

6. Reviewer 1: Although [the WHR] is still statistically lower than it men (now shown) as indicated in the reply, the biological significance is less clear and not addressed -- the apparently limited range of WHR /fat distribution in women may provide a less than optimally powered test of the hypothesis about body fat distribution as clearly ‘pear shape’ women (WHR < 0.75) are under-represented. This limitation is partially acknowledged but not explicitly made.

---Proposed response---

The mean difference of the WHR between men and women is 0.05. Although the biological significance between men and women was not often explicitly presented in literature as most of the studies, instead of interpreting the differences between two sexes, examine the distribution of the WHR within one sex. There are multiple WHR cutoff points for men and women regarding different disease outcomes such as diabetes, hypertension, dyslipidemia, etc. For instance, Obesity in Asia Collaboration initiative investigated using WHR as a predictor of the risk of hypertension, the paper recommended optimum range of WHR as 0.92 – 0.94 in men, 0.80 – 0.88 in women.² Therefore, a difference of 0.05 could potentially have larger biological

significance in men than women regarding risk of getting hypertension. However, the biological significance of the difference between the two sexes was not addressed. There are other WHR cutoff points like this for other related outcomes.³⁻⁶

As we continue to recruit participants, we hope we can recruit a more diverse population of WHR. Interestingly, low variation in the "explanatory" variable is likely to reduce the study's ability to detect an association. That we find an association is consistent with (though in no way definitive proof of) there being large associations in the population. The strongest way to assess the veracity of underlying association is for a prospective, designed, and well-powered study to follow-on to this study.

To address this limitation, we added more explicit wording in the limitation paragraph. (page 8, line 291 – 292)

Reference

1. Federation ID. *IDF Consensus Worldwide Definition of the Metabolic Syndrome*. International Diabetes Federation;2006.
2. Obesity in Asia C. Is central obesity a better discriminator of the risk of hypertension than body mass index in ethnically diverse populations? *J Hypertens*. 2008;26(2):169-177.
3. Ko GT, Chan JC, Cockram CS, Woo J. Prediction of hypertension, diabetes, dyslipidaemia or albuminuria using simple anthropometric indexes in Hong Kong Chinese. *Int J Obes Relat Metab Disord*. 1999;23(11):1136-1142.
4. Qiao Q, Nyamdorj R. The optimal cutoff values and their performance of waist circumference and waist-to-hip ratio for diagnosing type II diabetes. *Eur J Clin Nutr*. 2010;64(1):23-29.
5. Jia WP, Lu JX, Xiang KS, Bao YQ, Lu HJ, Chen L. Prediction of abdominal visceral obesity from body mass index, waist circumference and waist-hip ratio in Chinese adults: receiver operating characteristic curves analysis. *Biomedical and environmental sciences : BES*. 2003;16(3):206-211.
6. Lear SA, James PT, Ko GT, Kumanyika S. Appropriateness of waist circumference and waist-to-hip ratio cutoffs for different ethnic groups. *Eur J Clin Nutr*. 2010;64(1):42-61.

REVIEWERS' COMMENTS:

Reviewer #1 (Remarks to the Author):

Concerns were carefully addressed, but the 2 point main points were largely misunderstood. An attempt to clarify is offered below only for the authors information.

1. The authors took the term 'metabolic syndrome (MS)' too literally, and not as meant (apologies for lack of clarity).

Each metabolic syndrome characteristic is a continuous variable (e.g. glucose, lipids, BP, etc). So there is not reason to have to use the standard clinical definition for MS in the analysis.

Yes, Android fat ratio is a good marker for each of the metabolic syndrome characteristic, but the MS is a clinical and controversial concept. Considering specific elements of it make more biological sense because 2 individuals with the MS may have different elements of it.

2. The authors now state in the discussion that a relationship "between fat distribution and menopausal status was not detected in the female participants". This was not the question really.. the question is whether menopause status affects the relationship and microbiome outcomes in women (in a sex specific multivariate model) and on first principles one might expect this. The effect of menopause on fat distribution is not especially strong - so not surprising they don't detect in their relatively small and cross-sectional study.